# BARTSCORE:
# Evaluating Generated Text as Text Generation

**Weizhe Yuan**
Carnegie Mellon University
weizhey@cs.cmu.edu

**Graham Neubig**
Carnegie Mellon University
gneubig@cs.cmu.edu

**Pengfei Liu** *
Carnegie Mellon University
pliu3@cs.cmu.edu

## Abstract

A wide variety of NLP applications, such as machine translation, summarization, and dialog, involve text generation. One major challenge for these applications is how to *evaluate* whether such generated texts are actually fluent, accurate, or effective. In this work, we conceptualize the *evaluation of generated text as a text generation problem*, modeled using pre-trained sequence-to-sequence models. The general idea is that models trained to convert the generated text to/from a reference output or the source text will achieve higher scores when the generated text is better. We operationalize this idea using BART [32], an encoder-decoder based pre-trained model, and propose a metric BARTSCORE with a number of variants that can be flexibly applied in an unsupervised fashion to evaluation of text from different perspectives (e.g. informativeness, fluency, or factuality). BARTSCORE is conceptually simple and empirically effective. It can outperform existing top-scoring metrics in 16 of 22 test settings, covering evaluation of 16 datasets (e.g., machine translation, text summarization) and 7 different perspectives (e.g., informativeness, factuality). Code to calculate BARTScore is available at https://github.com/neulab/BARTScore, and we have released an interactive leaderboard for meta-evaluation at http://explainaboard.nlpedia.ai/leaderboard/task-meval/ on the EXPLAINABOARD platform [38], which allows us to interactively understand the strengths, weaknesses, and complementarity of each metric.

## 1 Introduction

One defining feature of recent NLP models is the use of neural representations trained on raw text, using unsupervised objectives such as language modeling [6,53], or denoising autoencoding [9,32,54]. By learning to predict the words or sentences in natural text, these models simultaneously learn to extract features that not only benefit mainstream NLP tasks such as information extraction [23,37], question answering [1,26], text summarization [40,77] but also have proven effective in development of automatic metrics for evaluation of text generation itself [62,65]. For example, BERTScore [75] and MoverScore [76] take features extracted by BERT [9] and apply unsupervised matching functions to compare system outputs against references. Other works build supervised frameworks that use the extracted features to learn to rank [56] or regress [62] to human evaluation scores.

However, in the context of generation evaluation, one may note that there is a decided *disconnect between how models are pre-trained using text generation objectives and how they are used as down-stream feature extractors*. This leads to potential under-utilization of the pre-trained model parameters. For example, the output prediction layer is not used at all in this case. This disconnect is particularly striking because of the close connection between the pre-training objectives and the generation tasks we want to evaluate.

---

* Corresponding author.

35th Conference on Neural Information Processing Systems (NeurIPS 2021).

In this paper, we instead argue for a formulation of *evaluation of generated text as a text generation problem*, directly evaluating text through the lens of its probability of being generated from or generating other textual inputs and outputs. This is a better match with the underlying pre-training tasks and allows us to more fully take advantage of the parameters learned during the pre-training phase. We solve the modeling problem with a pre-trained sequence-to-sequence (seq2seq) model, specifically BART [32], and devise a metric named BARTSCORE, which has the following characteristics: (1) BARTSCORE is parameter- and data-efficient. Architecturally there are no extra parameters beyond those used in pre-training itself, and it is an unsupervised metric that doesn't require human judgments to train. (2) BARTSCORE can better support evaluation of generated text from different perspectives (e.g., informativeness, coherence, factuality, §4) by adjusting the inputs and outputs of the conditional text generation problem, as we demonstrate in §3.2. This is in contrast to most previous work, which mostly examines correlation of the devised metrics with output quality from a limited number of perspectives. (3) BARTSCORE can be further enhanced by (i) providing textual prompts that bring the evaluation task closer to the pre-training task, or (ii) updating the underlying model by fine-tuning BART based on downstream generation tasks (e.g., text summarization).

Experimentally, we evaluate different variants of BARTSCORE from 7 perspectives on 16 datasets. BARTSCORE achieves the best performance in 16 of 22 test settings against existing top-scoring metrics. Empirical results also show the effectiveness of the *prompting* strategy supported by BARTSCORE. For example, simply adding the phrase "such as" to the translated text when using BARTSCORE can lead to a 3% point absolute improvement in correlation on "German-English" machine translation (MT) evaluation. Additional analysis shows that BARTSCORE is more robust when dealing with high-quality texts generated by top-performing systems.

## 2   Preliminaries

### 2.1   Problem Formulation

As stated above, our goal is to assess the quality of generated text [3, 46]. In this work, we focus on conditional text generation (e.g., machine translation), where the goal is to generate a *hypothesis* ($\boldsymbol{h} = h_1, \cdots, h_m$) based on a given *source* text ($\boldsymbol{s} = s_1, \cdots, s_n$). Commonly, one or multiple human-created *references* ($\boldsymbol{r} = r_1, \cdots, r_l$) are provided to aid this evaluation.

### 2.2   Gold-standard Human Evaluation

In general, the gold-standard method for evaluating such texts is still human evaluation, where human annotators assess the generated texts' quality. This evaluation can be done from perspectives, and we list a few common varieties below (all are investigated in §4):

1. **Informativeness** (INFO): How well the generated hypothesis captures the key ideas of the source text [18].
2. **Relevance** (REL): How consistent the generated hypothesis is with respect to the source text [19].
3. **Fluency** (FLU): Whether the text has no formatting problems, capitalization errors or obviously ungrammatical sentences (e.g., fragments, missing components) that make the text difficult to read [13].
4. **Coherence** (COH): Whether the text builds from sentence to sentence to a coherent body of information about a topic [7].
5. **Factuality** (FAC): Whether the generated hypothesis contains only statements entailed by the source text [30].
6. **Semantic Coverage** (COV): How many semantic content units from reference texts are covered by the generated hypothesis [49].
7. **Adequacy** (ADE): Whether the output conveys the same meaning as the input sentence, and none of the message is lost, added, or distorted [29].

Most existing evaluation metrics were designed to cover a small subset of these perspectives. For example, BLEU [50] aims to capture the adequacy and fluency of translations, while ROUGE [36] was designed to match the semantic coverage metric. Some metrics, particularly trainable ones, can perform evaluation from different perspectives but generally require maximizing correlation with each type of judgment separately [8].

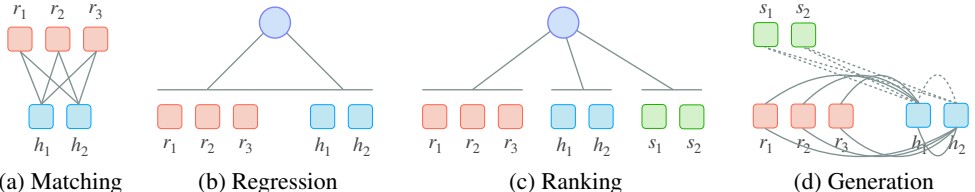

Figure 1: Evaluation metrics as different tasks, where $s_i$, $h_i$ and $r_j$ represent *source*, *hypothesis* and *reference* words respectively.

As we describe more in §4, BARTSCORE can evaluate text from the great majority of these perspectives, significantly expanding its applicability compared to these metrics.

## 2.3 Evaluation as Different Tasks

There is a recent trend that leverages neural models for automated evaluation in different ways, as shown in Fig. 1. We first elaborate on their characteristics by highlighting differences in task formulation and evaluation perspectives.

**T1: Unsupervised Matching.** Unsupervised matching metrics aim to measure the semantic equivalence between the reference and hypothesis by using a *token-level matching* functions in distributed representation space, such as BERTScore [75], MoverScore [76] or discrete string space like ROUGE [35], BLEU [50], CHRF [52]. Although similar matching functions can be used to assess the quality beyond semantic equivalence (e.g, factuality, a relationship between source text and hypothesis), to our knowledge prior research has not attested to the capability of unsupervised matching methods in this regard; we explore this further in our experiments (Tab. 5).

**T2: Supervised Regression.** Regression-based models introduce a parameterized regression layer, which would be learned in a supervised fashion to accurately predict human judgments. Examples include recent metrics BLEURT [62], COMET [56] and traditional metrics like $S^3$ [51], VRM [21].

**T3: Supervised Ranking.** Evaluation can also be conceived as a ranking problem, where the main idea is to learn a scoring function that assigns a higher score to better hypotheses than to worse ones. Examples include COMET [56] and BEER [64], where COMET focuses the machine translation task and relies on human judgments to tune parameters in ranking or regression layers, and BEER combines many simple features in a tunable linear model of MT evaluation metrics.

**T4: Text Generation.** In this work, we formulate evaluating generated text as a text generation task from pre-trained language models. The basic idea is that a high-quality hypothesis will be easily generated based on source or reference text or vice-versa. This has not been covered as extensively in previous work, with one notable exception being PRISM [65]. Our work differs from PRISM in several ways: (i) PRISM formulates evaluation as a paraphrasing task, whose definition that two texts are with the same meaning limits its applicable scenarios, like factuality evaluation in text summarization that takes source documents and generated summaries as input whose semantic space are different. (ii) PRISM trained a model from scratch on parallel data while BARTSCORE is based on open-sourced pre-trained seq2seq models. (iii) BARTSCORE supports prompt-based learning [59, 63] which hasn't been examined in PRISM.

## 3 BARTScore

### 3.1 Sequence-to-Sequence Pre-trained Models

Although pre-trained models differ along different axes, one of the main axes of variation is the training objective, with two main variants: language modeling objectives (e.g., masked language modeling [9]) and seq2seq objectives [54]. In particular, seq2seq pre-trained models are particularly well-suited to conditioned generation tasks since they consist of both an encoder and a decoder, and predictions are made auto-regressively [32]. In this work, we operationalize our idea by using

BART [32] as our backbone due to its superior performance in text generation [12, 42, 71]. We also report preliminary experiments comparing BART with T5 [54] and PEGASUS [74] in the Appendix.

Given a seq2seq model parameterized by $\theta$, a source sequence containing $n$ tokens $\mathbf{x} = \{x_1, \cdots, x_n\}$ and a target sequence containing $m$ tokens $\mathbf{y} = \{y_1, \cdots, y_m\}$. We can factorize the generation probability of $\mathbf{y}$ conditioned on $\mathbf{x}$ as follows:

$$p(\mathbf{y}|\mathbf{x}, \theta) = \prod_{t=1}^{m} p(\mathbf{y}_t|\mathbf{y}_{<t}, \mathbf{x}, \theta) \tag{1}$$

By exploring these probabilities, we design metrics that can gauge the quality of the generated text.

## 3.2 BARTScore

The most general form of our proposed BARTSCORE is shown in Eq. 2, where we use the weighted log probability of one text $\mathbf{y}$ given another text $\mathbf{x}$. The weights are used to put different emphasis on different tokens, which can be instantiated using different methods like Inverse Document Frequency (IDF) [25] etc. In our work, we weigh each token equally.[2]

$$\text{BARTSCORE} = \sum_{t=1}^{m} \omega_t \log p(\mathbf{y}_t|\mathbf{y}_{<t}, \mathbf{x}, \theta) \tag{2}$$

Due to its generation task-based formulation and ability to utilize the entirety of BART's pre-trained parameters, BARTSCORE can be flexibly used in different evaluation scenarios. We specifically present four methods for using BARTSCORE based on different generation directions, which are,

- *Faithfulness* ($s \rightarrow h$): from source document to hypothesis $p(h|s, \theta)$. This direction measures how likely it is that the hypothesis could be generated based on the source text. Potential application scenarios are factuality and relevance introduced in §2.2. This measure can also be used for estimating measures of the quality of only the target text, such as coherence and fluency (§2.2).
- *Precision* ($r \rightarrow h$): from reference text to system-generated text $p(h|r, \theta)$. This direction assesses how likely the hypothesis could be constructed based on the gold reference and is suitable for the precision-focused scenario.
- *Recall* ($h \rightarrow r$): from system-generated text to reference text $p(r|h, \theta)$. This version quantifies how easily a gold reference could be generated by the hypothesis and is suitable for pyramid-based evaluation (i.e., semantic coverage introduced in §2.2) in summarization task since pyramid score measures fine-grained Semantic Content Units (SCUs) [49] covered by system-generated texts.
- $\mathcal{F}$ score ($r \leftrightarrow h$): Consider both directions and use the arithmetic average of *Precision* and *Recall* ones. This version can be broadly used to evaluate the semantic overlap (informativeness, adequacy detailed in §2.2) between reference texts and generated texts.

## 3.3 BARTScore Variants

We also investigate two extensions to BARTSCORE: (i) changing $\mathbf{x}$ and $\mathbf{y}$ through prompting, which can *bring the evaluation task closer to the pre-training task*. (ii) changing $\theta$ by considering different fine-tuning tasks, which can *bring the pre-training domain closer to the evaluation task*.

### 3.3.1 Prompt

*Prompting* is a practice of adding short phrases to the input or output to encourage pre-trained models to perform specific tasks, which has been proven effective in several other NLP scenarios [24, 57, 58, 60, 63]. The generative formulation of BARTSCORE makes it relatively easy to incorporate these insights here as well; we name this variant BARTSCORE-PROMPT.

Given a prompt of $l$ tokens $\mathbf{z} = \{z_1, \cdots, z_l\}$, we can either (i) append it to the source text, in which case we get $\mathbf{x}' = \{x_1, \cdots, x_n, z_1, \cdots, z_l\}$, and calculate the score based on this new source text using Eq.2. or (ii) prepend it to the target text, getting $\mathbf{y}' = \{z_1, \cdots, z_l, y_1, \cdots, y_m\}$. Then we can also use Eq.2 given the new target text.

---

[2]We have tried several other weighting schemes, including: (i) uniform weighting while ignoring stop words. (ii) IDF weighting. (iii) using the prior probability of each target token (calculated within the target sequence) as the weighting factor. However, none of those outperformed the uniform weighting scheme.

### 3.3.2 Fine-tuning Task

Different from BERT-based metrics, which typically use classification-based tasks (e.g., natural language inference) [67] to fine-tune, BARTSCORE can be fine-tuned using generation-based tasks, which will make the pre-training domain closer to the evaluation task. In this paper, we explore two downstream tasks. (1) Summarization. We use BART fine-tuned on `CNNDM` dataset [20], which is available off-the-shelf in Huggingface Transformers [70]. (2) Paraphrasing. We continue fine-tuning BART from (1) on `ParaBank2` dataset [22], which contains a large paraphrase collection. We used a random subset of 30,000 data and fine-tuned for one epoch with a batch size of 20 and a learning rate of $5e^{-5}$. We used two 2080Ti GPUs, and the training time is less than one hour.

## 4 Experiment

This section aims to evaluate the reliability of different automated metrics, which is commonly achieved by quantifying how well different metrics correlate with human judgments using measures (e.g., Spearman Correlation [72]) defined below (§4.1.2).

### 4.1 Baselines and Datasets

#### 4.1.1 Evaluation Metrics

We comprehensively examine metrics outlined in §2.3, which either require human judgments to train (i.e., *supervised metrics*): **COMET** [56], **BLEURT** [62], or are human judgment-free (i.e., *unsupervised*): **BLEU** [50] **ROUGE-1 and ROUGE-2, ROUGE-L**, **CHRF** [52], **PRISM** [65], **MoverScore** [76], **BERTScore** [75]. The detailed comparisons of those metrics can be found in Appendix. We use the official code for each metric.

#### 4.1.2 Measures for Meta Evaluation

**Pearson Correlation** [15] measures the linear correlation between two sets of data. **Spearman Correlation** [72] assesses the monotonic relationships between two variables. **Kendall's Tau** [27] measures the ordinal association between two measured quantities. **Accuracy**, in our experiments, measures the percentage of correct ranking between factual texts and non-factual texts. We follow previous works in the choices of measures for different datasets to make a fair comparison.

#### 4.1.3 Datasets

The datasets we use are summarized in Tab. 1. We consider three different tasks: summarization (SUM), machine translation (MT), and data-to-text (D2T).

**Machine Translation** We obtain the source language sentences, machine-translated texts and reference texts from the `WMT19` metrics shared task [44]. We use the DARR corpus and consider 7 language pairs, which are `de-en`, `fi-en`, `gu-en`, `kk-en`, `lt-en`, `ru-en`, `zh-en`.

**Text Summarization** (1) `REALSumm` [4] is a meta-evaluation dataset for text summarization which measures *pyramid recall* of each system-generated summary. (2) `SummEval` [13] is a collection of human judgments of model-generated summaries on the `CNNDM` dataset annotated by both expert judges and crowd-source workers. Each system generated summary is gauged through the lens of *coherence*, *consistency*, *fluency* and *relevance*.[3] (3) `NeR18` The

Table 1: A summary of tasks, datasets, and evaluation perspectives that we have covered in our experiments. Explanation of evaluation perspectives can be found in §2.2.

| Tasks | Datasets | Eval. Perspectives |
|---|---|---|
| SUM | REALSUM | Cov |
| | SummEval | COH FAC FLU INFO |
| | NeR18 | COH FLU REL INFO |
| | Rank19 QAGS-C QAGS-X | FAC |
| MT | DE FI GU KK IT RU ZH | ADE FLU |
| D2T | BAGEL SFHOT SFRES | INFO |

---

[3]We rephrase the original "relevance" into "informativeness" and "consistency" into "factuality" based on the descriptions in their paper and our definitions in §2.2.

`NEWSROOM` dataset [18] contains 60 articles with summaries generated by 7 different methods are annotated with human scores in terms of *coherence*, *fluency*, *informativeness*, *relevance*.

**Factuality** (1) `Rank19` [14] is used to meta-evaluate factuality metrics. It is a collection of 373 triples of a source sentence with two summary sentences, one correct and one incorrect. (2) `QAGS20` [66] collected 235 test outputs on `CNNDM` dataset from [16] and 239 test outputs on `XSUM` dataset [47] from BART fine-tuned on `XSUM`. Sentences in each summary are annotated with correctness scores w.r.t. factuality.

**Data to Text** We consider the following datasets which target utterance generation for spoken dialogue systems. (1) `BAGEL` [45] provides information about restaurants. (2) `SFHOT` [69] provides information about hotels in San Francisco. (3) `SFRES` [69] provides information about restaurants in San Francisco. They contain 202, 398, and 581 samples respectively, each sample consists of one meaning representation, multiple references, and utterances generated by different systems.

## 4.2 Setup

### 4.2.1 Prompt Design

To perform prompting, we first need to find proper prompts within a search space. Instead of considering a large discrete search space [63][4] or continuous search space [34], we use simple heuristics to narrow our search space. In particular, we use manually devised seed prompts and gather paraphrases to construct our prompt set.[5] The seed prompts and some examples of paraphrased prompts are shown in Tab. 2. Details are listed in the Appendix.

Table 2: Seed prompts and examples of final prompts. "Number" denotes the size of our final prompt set that was acquired from the seed prompts.

| Usage | Number | Seed | Example |
|---|---|---|---|
| $s \rightarrow h$ | 70 | in summary | in short, in a word, to sum up |
| $h \leftrightarrow r$ | 34 | in other words | to rephrase it, that is to say, i.e. |

### 4.2.2 Settings

**Variants.** We consider four variants of BARTSCORE, which are (1) BARTSCORE, which uses the vanilla BART; (2) BARTSCORE-CNN, which uses the BART fine-tuned on the summarization dataset `CNNDM`; (3) BARTSCORE-CNN-PARA, where BART is first fine-tuned on `CNNDM`, then fine-tuned on `ParaBank2`. (4) BARTSCORE-PROMPT, which is enhanced by adding prompts.

**Selection of Prompts.** For the summarization and data-to-text tasks, we use all entries (either all prompts designed for $s \rightarrow h$ or all prompts designed for $h \leftrightarrow r$ depending on the BARTScore usage chosen) in the prompt set by prefixing the decoder input and getting different generation scores (calculated by Eq.2) for each hypothesis based on different prompts. We finally get the score for one hypothesis by taking the average of all its generation scores using different prompts ( [24]; details about *prompt ensembling* can be found in the Appendix). For the machine translation task, due to the more expensive computational cost brought by larger text sets, we first use WMT18 [43] as a development set to search for one best prompt and obtain the phrase "Such as", which is then used for the test language pairs.

**Selection of BARTScore Usage.** Although BARTSCORE can be used in different ways (shown in §3.2)), in different tasks, they can be chosen based on how targeted evaluation perspectives are defined (described in §2.2) as well as the types of tasks. Specifically, (i) For those datasets whose gold standard human evaluation are obtained based on recall-based pyramid method, we adopt recall-based BARTSCORE ($h \rightarrow r$). (ii) For those datasets whose human judgments focus on linguistic quality (coherence, fluency) and factual correctness (factuality), or the source and hypothesis texts are in the same modality (i.e., language), we use faithfulness-based BARTSCORE ($s \rightarrow h$). (iii) For data-to-text and machine translation tasks, to make a fair comparison, we use BARTSCORE with the F-score version that other existing works [65] have adopted when evaluating generated texts.

---

[4]We explored this first and found that discovered prompts led to worse performance.
[5]We use the website https://www.wordhippo.com/ to search for synonyms.

Table 3: Kendall's Tau correlation of different metrics on WMT19 dataset. The highest correlation for each language pair achieved by *unsupervised* method is **bold**, and the highest correlation *overall* is underlined. **Avg.** denotes the average correlation achieved by a metric across all language pairs.

| | de-en | fi-en | gu-en | kk-en | lt-en | ru-en | zh-en | Avg. |
|---|---|---|---|---|---|---|---|---|
| SUPERVISED METHODS | | | | | | | | |
| BLEURT | 0.174 | 0.374 | 0.313 | 0.372 | 0.388 | 0.220 | 0.436 | 0.325 |
| COMET | 0.219 | 0.369 | 0.316 | 0.378 | 0.405 | 0.226 | 0.462 | 0.339 |
| UNSUPERVISED METHODS | | | | | | | | |
| BLEU | 0.054 | 0.236 | 0.194 | 0.276 | 0.249 | 0.115 | 0.321 | 0.206 |
| CHRF | 0.123 | 0.292 | 0.240 | 0.323 | 0.304 | 0.177 | 0.371 | 0.261 |
| PRISM | 0.199 | 0.366 | **0.320** | 0.362 | 0.382 | 0.220 | 0.434 | 0.326 |
| BERTScore | 0.190 | 0.354 | 0.292 | 0.351 | 0.381 | **0.221** | 0.430 | 0.317 |
| BARTSCORE | 0.156 | 0.335 | 0.273 | 0.324 | 0.322 | 0.167 | 0.389 | 0.281 |
| + CNN | 0.190 | 0.365 | 0.300 | 0.348 | 0.384 | 0.208 | 0.425 | 0.317 |
| + CNN + Para | 0.205† | 0.370† | 0.316 | **0.378**† | 0.386† | 0.219 | 0.442† | 0.331 |
| + CNN + Para + Prompt | **0.238**‡ | **0.374**‡ | 0.318 | 0.376† | 0.386† | 0.219 | **0.447**‡ | **0.337** |

Table 4: Spearman correlation of different metrics on three human judgement datasets. For prompt-based learning, we consider adding prompts to the best-performing BARTSCORE ($\Omega$) on each dataset. The highest correlation overall for each aspect on each dataset is **bold**.

| | REALSumm | SummEval | | | | NeR18 | | | | |
|---|---|---|---|---|---|---|---|---|---|---|
| | COV | COH | FAC | FLU | INFO | COH | FLU | INFO | REL | Avg. |
| ROUGE-1 | **0.498** | 0.167 | 0.160 | 0.115 | 0.326 | 0.095 | 0.104 | 0.130 | 0.147 | 0.194 |
| ROUGE-2 | 0.423 | 0.184 | 0.187 | 0.159 | 0.290 | 0.026 | 0.048 | 0.079 | 0.091 | 0.165 |
| ROUGE-L | 0.488 | 0.128 | 0.115 | 0.105 | 0.311 | 0.064 | 0.072 | 0.089 | 0.106 | 0.164 |
| BERTScore | 0.440 | 0.284 | 0.110 | 0.193 | 0.312 | 0.147 | 0.170 | 0.131 | 0.163 | 0.217 |
| MoverScore | 0.372 | 0.159 | 0.157 | 0.129 | 0.318 | 0.161 | 0.120 | 0.188 | 0.195 | 0.200 |
| PRISM | 0.411 | 0.249 | 0.345 | 0.254 | 0.212 | 0.573 | 0.532 | 0.561 | 0.553 | 0.410 |
| BARTSCORE | 0.441 | 0.322† | 0.311 | 0.248 | 0.264 | 0.679† | 0.670† | 0.646† | 0.604† | 0.465 |
| + CNN | 0.475 | **0.448**‡ | 0.382† | 0.356† | 0.356† | 0.653† | 0.640† | 0.616† | 0.567 | 0.499 |
| + CNN + Para | 0.471 | 0.424† | **0.401**‡ | **0.378**‡ | 0.313 | 0.657† | 0.652† | 0.614† | 0.562 | 0.497 |
| + $\Omega$ + Prompt | 0.488 | 0.407† | 0.378† | 0.338† | **0.368**‡ | **0.701**‡ | **0.679**‡ | **0.686**‡ | **0.620**‡ | **0.518** |

**Significance Tests.** To perform rigorous analysis, we adopt the bootstrapping method (p-value < 0.05) [28] for pair-wise significance tests. In all tables, we use † on BARTSCORE if it significantly ($p < 0.05$) outperforms other unsupervised metrics *excluding* BARTSCORE variants. We use ‡ on BARTSCORE if it significantly outperforms all other unsupervised metrics *including* BARTSCORE variants.

## 4.3 Experimental Results

### 4.3.1 Machine Translation

Tab. 3 illustrates Kendall's Tau correlation of diverse metrics on different language pairs. We can observe that: (1) BARTSCORE enhanced by fine-tuning tasks (CNN+Para) can significantly outperform all other unsupervised methods on five language pairs and achieve comparable results on the other two. (2) The performance of BARTSCORE can be further improved by simply adding a prompt (i.e., such as) without any other overhead. Notably, on the language pair de-en, using the prompt results in a 0.033 improvement, which even significantly surpasses existing state-of-the-art supervised metrics BLEURT and COMET. This suggests a promising future direction for metric design: *searching for proper prompts to better leverage knowledge stored in pre-trained language models instead of training on human judgment data* [31].

### 4.3.2 Text Summarization

Tab. 4 shows the meta-evaluation results of different metrics on the summarization task. We can observe that: (1) Simply vanilla BARTSCORE can outperform BERTScore and MoverScore by a large margin on 8 settings except the INFO perspective on SummEval. Strikingly, it achieves improvements of 0.251 and 0.265 over BERTScore and MoverScore respectively. (2) The improvement on REALSum and SummEval datasets can be further improved when introducing fine-tuning tasks. However, fine-tuning does not improve on the NeR18 dataset, likely because this dataset only contains 7 systems with easily distinguishable quality, and vanilla BARTSCORE can already achieve a high level of correlation ($> 0.6$ on average). (3) Our prompt combination strategy can consistently improve the performance on *informativeness*, up to 0.072 Spearman correlation on the NeR18 dataset

and 0.055 on SummEval. However, the performance from other perspectives such as *fluency* and *factuality* do not show consistent improvements, which we will elaborate on later (§4.4.2).

**Analysis on Factuality Datasets** The goal of these datasets is to judge whether a *short generated summary* is faithful to the original *long documents*. As shown in Tab. 5, we observe that (1) BARTSCORE + CNN can almost match *human baseline* on Rank19 and outperform all other metrics, including the most recent top-performing factuality metrics FactCC and QAGS by a large margin. (2) Using paraphrase as a fine-tuning task will reduce BARTSCORE's performance, which is reasonable since these two texts (i.e., the summary and document) shouldn't maintain the paraphrased relationship in general. (3) Introducing prompts does not bring an improvement, even resulting in a performance decrease.

Table 5: Results on Rank19 and QAGS datasets. where "Q" represents QAGS. Metrics achieve highest correlation are **bold**.

|  | Rank19 | Q-CNN | Q-XSUM |
|---|---|---|---|
|  | Acc. | Pearson | |
| ROUGE-1 | 0.568 | 0.338 | -0.008 |
| ROUGE-2 | 0.630 | 0.459 | 0.097 |
| ROUGE-L | 0.587 | 0.357 | 0.024 |
| BERTScore | 0.713 | 0.576 | 0.024 |
| MoverScore | 0.713 | 0.414 | 0.054 |
| PRISM | 0.780 | 0.479 | 0.025 |
| FactCC [30] | 0.700 | – | – |
| QAGS [66] | 0.721 | 0.545 | 0.175 |
| Human [14] | 0.839 | – | – |
| BARTSCORE | 0.684 | 0.661† | 0.009 |
| + CNN | **0.836‡** | **0.735‡** | **0.184‡** |
| + CNN + Para | 0.788 | 0.680† | 0.074 |
| + CNN + Prompt | 0.796 | 0.719† | 0.094 |

### 4.3.3 Data-to-text

The experiment results on data-to-text datasets are shown in Tab. 6. We observe that (1) fine-tuning on the CNNDM dataset can consistently boost the correlation, for example, up to 0.056 gain on BAGEL. (2) Additionally, further fine-tuning on paraphrase datasets results in even higher performance compared to the version without any fine-tuning, up to 0.083 Spearman correlation on BAGEL dataset. These results surpass all existing top-performing metrics. (3) Our proposed prompt combination strategy can consistently improve correlation, on average 0.028 Spearman correlation. This is consistent with the findings in §4.3.2 that we can improve the aspect of *informativeness* through proper prompting.

Table 6: Results on data-to-text datasets. We report Spearman correlation. Metrics achieve highest correlation are **bold**.

|  | BAGEL | SFRES | SFHOT | Avg. |
|---|---|---|---|---|
| ROUGE-1 | 0.234 | 0.115 | 0.118 | 0.156 |
| ROUGE-2 | 0.199 | 0.116 | 0.088 | 0.134 |
| ROUGE-L | 0.189 | 0.103 | 0.110 | 0.134 |
| BERTScore | 0.289 | 0.156 | 0.135 | 0.193 |
| MoverScore | 0.284 | 0.153 | 0.172 | 0.203 |
| PRISM | 0.305 | 0.155 | 0.196 | 0.219 |
| BARTSCORE | 0.247 | 0.164† | 0.158 | 0.190 |
| + CNN | 0.303 | 0.191† | 0.190 | 0.228 |
| + CNN + Para | 0.330† | 0.185† | 0.211† | 0.242 |
| + Ω + Prompt | **0.336‡** | **0.238‡** | **0.235‡** | **0.270** |

### 4.4 Analysis

We design experiments to better understand the mechanism by which BARTSCORE obtains these promising results, specifically asking three questions: Q1: Compared to other unsupervised metrics, where does BARTSCORE outperform them? Q2: How does adding prompts benefit evaluation? Q3: Will BARTScore introduce biases in unpredictable ways?

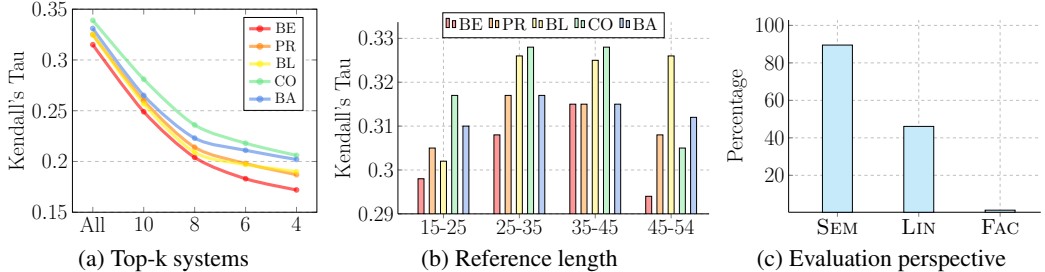

Figure 2: Fine-grained analysis (a,b) and prompt analysis (c). In (a, b), BE, PR, BL, CO, BA represent BERTScore, PRISM, BLEURT, COMET and BARTSCORE respectively. In (c), SEM, LIN, FAC denote *semantic overlap*, *linguistic quality* and *factual correctness* respectively.

### 4.4.1 Fine-grained Analysis

To answer Q1, we choose the MT task and break down the performance of each metric into different buckets based on different axes.

**Top-k Systems** We report the average correlation across all language pairs achieved by each metric given only translations from top-$k$ systems. We vary the number of $k$, and the results are shown in Fig. 2-(a). We can see that BARTSCORE can outperform all other metrics (including one supervised metric BLEURT) except the existing state-of-the-art supervised metric COMET for different $k$, and the decrease in correlation becomes smoother than others when considering top-scoring systems. This indicates that BARTSCORE is robust to high-quality generated texts.

**Reference Length** We break down each test set into four buckets based on the reference length, which are $[15, 25)$, $[25, 35)$, $[35, 45)$, $[45, 54]$ and compute the Kendall's Tau average correlation of different metrics across all language pairs within each bucket.[6] The results are shown in Fig. 2-(b). We observe that BARTSCORE can outperform or tie with other unsupervised metrics over different reference lengths. Also, its correlation with human judgments is more stable compared to all other metrics. This indicates its robustness to different input lengths. More other analyses can be found in Appendix.

### 4.4.2 Prompt Analysis

For Q2, we choose the summarization and data-to-text tasks for analysis where we used all prompts from our prompt set. We first group all the evaluation perspectives into three categories: (1) *semantic overlap* (informativeness, pyramid score, and relevance) (2) *linguistic quality* (fluency, coherence) (3) *factual correctness* (factuality). We then calculate the percentage of prompts that result in performance improvements for each perspective within a dataset. Finally, we compute the average percentage of prompts that can lead to performance gains for each category. The results are shown in Tab. 2-(c). We can see that for *semantic overlap*, almost all prompts can lead to the performance increase, while for *factuality* only a few prompts can improve the performance. This also explains the results in §4.3.2 where we found that *combining the results of different prompts can lead to consistent increases in semantic overlap but worse performance in factuality*. Regarding *linguistic quality*, the effect of adding a prompt is not that predictive, which is also consistent with our findings in §4.3.2.

### 4.4.3 Bias Analysis

To answer Q3, we conduct bias analysis. Bias would indicate that the scores are too high or too low compared to the scores they are given by human annotators. Therefore, to see whether such biases exist, we inspected the rank differences given by human annotators and BARTScore (fine-tuned on CNNDM dataset) on the REALSumm dataset where 24 systems are considered, including both abstractive models and extractive models as well as models based on pre-trained models and models

---

[6]In each bucket, we remove the language pairs that do not contain over 500 samples. This results in the removal of kk-en in $[35, 45)$ and the removal of gu-en, kk-en, lt-en in $[45, 54]$.

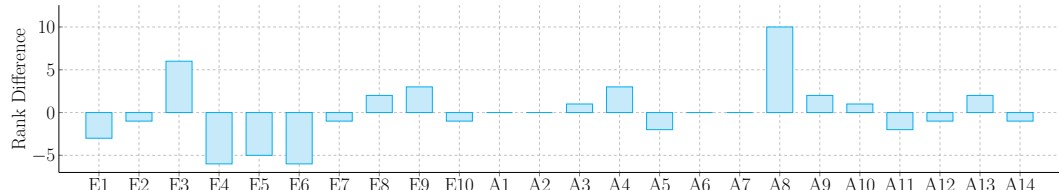

Figure 3: Bias analysis of BARTScore. The "Rank Difference" is the rank obtained using human judgements minus the rank got from BARTScore. Systems beginning with letter "E" are extractive systems while systems beginning with letter "A" are abstractive systems.

that are trained from scratch. We list all the systems below. And the resulting rank difference is shown in Fig. 3.

**Extractive Systems**  E1: BanditSum [11]; E2: Refresh [48]; E3: NeuSum [80]; E4: LSTM-PN-RL [79]; E5: BERT-TF-SL [79]; E6: BERT-TF-PN [79]; E7: BERT-LSTM-PN-RL [79]; E8: BERT-LSTM-PN [79]; E9: HeterGraph [68]; E10: MatchSum [78].

**Abstractive Systems**  A1: Ptr-Gen [61]; A2: Bottom-up [17]; A3: Fast-Abs-RL [5]; A4: Two-stage-RL [73]; A5: BERT-Ext-Abs [41]; A6: BERT-Abs [41]; A7: Trans-Abs [41]; A8: UniLM-1 [10]; A9: UniLM-2 [2]; A10: T5-base [55]; A11: T5-large [55]; A12: T5-11B [55]; A13: BART [33]; A14: SemSim [42].

As shown in Fig. 3, BARTScore is less effective at distinguishing the quality of extractive summarization systems while much better at distinguishing the quality of abstractive summarization systems. However, given that there is a trend for using abstractive systems as more and more pre-trained sequence-to-sequence models being proposed, BARTScore's weaknesses on extractive systems will be mitigated.

## 5   Implications and Future Directions

In this paper, we proposed a metric BARTSCORE that formulates evaluation of generated text as a text generation task, and empirically demonstrated its efficacy. Without the supervision of human judgments, BARTSCORE can effectively evaluate texts from 7 perspectives and achieve the best performance on 16 of 22 settings against existing top-scoring metrics. We highlight potential future directions based on what we have learned.

**Prompt-augmented metrics**  As an easy-to-use but powerful method, *prompting* [39] has achieved impressive performance particularly on semantic overlap-based evaluation perspectives. However, its effectiveness in factuality and linguistic quality-based perspectives has not been fully demonstrated in this paper. In the future, more works can explore how to make better use of prompts for these and other evaluation scenarios.

**Co-evolving evaluation metrics and systems**  BARTSCORE builds the connection between metric design and system design, which allows them to share their technological advances, thereby progressing together. For example, a better BART-based summarization *system* may be directly used as a more reliable automated *metric* for evaluating summaries, and this work makes them connected.

## Acknowledgments

The authors would like to thank the anonymous reviewers for their insightful comments and suggestions. The authors also thank Wei Zhao for assisting with reproducing baseline results. This work was supported by the Air Force Research Laboratory under agreement number FA8750-19-2-0200. The U.S. Government is authorized to reproduce and distribute reprints for Governmental purposes notwithstanding any copyright notation thereon. The views and conclusions contained herein are those of the authors and should not be interpreted as necessarily representing the official policies or endorsements, either expressed or implied, of the Air Force Research Laboratory or the U.S. Government.

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
