# A  Appendix

## A.1  Summary of Commonly Used Metrics for Text Generation

Table 1: Summary of commonly used metrics for text generation. $(S, H)$ represents whether a metric has a setting that uses source text and hypothesis text. $(R, H)$ denotes whether a metric has a setting that uses reference text and hypothesis text. $(S, R, H)$ indicates whether a metric has a setting that uses source text, hypothesis text and reference text. We use the following abbreviations for different tasks: SUM - Summarization, MT - Machine Translation, MUL - Multiple tasks, FAC - Factuality. For settings and tasks, we only list the ones justified by the original paper for each metric.

| Metrics | Supervised | Paradigm | $(S, H)$ | $(R, H)$ | $(S, R, H)$ | Task | Support FAC |
|---|---|---|---|---|---|---|---|
| ROUGE | ✗ | Match | | ✓ | | SUM | ✗ |
| BLEU | ✗ | Match | | ✓ | | MT | ✗ |
| CHRF | ✗ | Match | | ✓ | | MT | ✗ |
| BERTScore | ✗ | Match | | ✓ | | MUL | ✗ |
| MoverScore | ✗ | Match | | ✓ | | MUL | ✗ |
| PRISM | ✗ | Paraphrase | ✓ | ✓ | | MT | ✗ |
| BLEURT | ✓ | Regress | | ✓ | | MT | ✗ |
| S3 | ✓ | Regress | | | ✓ | SUM | ✗ |
| VRM | ✓ | Regress | | ✓ | | SUM | ✗ |
| COMET | ✓ | Regress, Rank | | | ✓ | MT | ✗ |
| BEER | ✓ | Rank | | ✓ | | MT | ✗ |
| BARTScore | ✗ | Generation | ✓ | ✓ | | MUL | ✓ |

## A.2  Pre-trained Model Selection

Besides BART, we also tried T5 and PEGASUS as our sequence-to-sequence model to get generation scores. We conduct experiments on WMT19, and the results are shown in Tab. 2. We don't observe improvements in using PEGASUS or T5 over BART.

Table 2: Experiment results for PEGASUS and T5 on the WMT19 dataset. The highest correlations are bold.

| | de-en | fi-en | gu-en | kk-en | lt-en | ru-en | zh-en |
|---|---|---|---|---|---|---|---|
| PEGASUS-large | 0.124 | 0.297 | 0.237 | 0.205 | 0.252 | 0.148 | 0.311 |
| PEGASUS-large-cnn | 0.174 | 0.361 | 0.297 | 0.337 | 0.373 | 0.215 | 0.415 |
| T5-base | 0.170 | 0.357 | **0.300** | 0.339 | 0.348 | **0.208** | 0.378 |
| T5-large | 0.168 | 0.353 | 0.287 | 0.332 | 0.335 | 0.193 | 0.383 |
| T5-base-cnn | 0.177 | 0.364 | 0.295 | 0.342 | 0.347 | 0.207 | 0.402 |
| BART | 0.156 | 0.335 | 0.273 | 0.324 | 0.322 | 0.167 | 0.389 |
| BART-cnn | **0.190** | **0.365** | **0.300** | **0.348** | **0.384** | **0.208** | **0.425** |

## A.3  Prompt Set

In Tab. 3, we list the full prompt set for both $s \to h$ direction and $h \leftrightarrow r$ direction.

## A.4  Prompt Combination

Given a source sequence $\mathbf{x}$, a target sequence $\mathbf{y}$ and a set of prompts $\mathbf{z}_1, \mathbf{z}_2, \cdots \mathbf{z}_n$. We denote the prompted target sequence as $[\mathbf{y} : \mathbf{z}_i]$ for any prompt $\mathbf{z}_i$. Under the sequence-to-sequence model

Table 3: Full prompt set for both $s \rightarrow h$ and $h \leftrightarrow r$

| | Prompt Set | | | | |
|---|---|---|---|---|---|
| $s \rightarrow h$ | Last | Tersely | Succinctly | In summation | To put it succinctly |
| | After | In brief | All in all | To summarize | Bringing up the rear |
| | Behind | In short | In outline | In a nutshell | To come to the point |
| | Lastly | Concisely | In closing | In conclusion | In the final analysis |
| | In sum | In precis | In passing | In winding up | Without wasting words |
| | To end | In a word | To conclude | Last in order | At the end of the day |
| | Curtly | Compactly | Summarising | In a few words | Without waste of words |
| | Crisply | Summarily | In the rear | As a final point | Finally yet importantly |
| | At last | To sum up | Summarizing | Not least of all | To put it in a nutshell |
| | Pithily | Basically | Laconically | To put it briefly | When all is said and done |
| | Shortly | In the end | At the rear | Not to mince words | To cut a long story short |
| | In fine | At the end | To be brief | Last but not least | Not to beat about the bush |
| | Finally | In essence | Last of all | Just as importantly | In drawing things to a close |
| | Briefly | Ultimately | Elliptically | To put it concisely | Not to put too fine a point on it |
| $h \leftrightarrow r$ | As | To wit | As it were | Case in point | As an illustration |
| | sc. | That is | Especially | That is to say | To give an example |
| | i.e. | Such as | For example | To rephrase it | To give an instance |
| | Like | Scilicet | Particularly | To be specific | To put it another way |
| | Viz. | Videlicet | Specifically | In plain English | By way of explanation |
| | Namely | Expressly | For instance | Take for example | By way of illustration |
| | id est | Specially | To illustrate | Strictly speaking | |

parameterized by $\theta$, we combine the generation scores using different prompts as follows:

$$\text{BARTSCORE-PROMPT} = \frac{1}{n} \sum_{i=1}^{n} \frac{1}{m_i} \sum_{t=1}^{m_i} \log p([\mathbf{y} : \mathbf{z}_i]_t | [\mathbf{y} : \mathbf{z}_i]_{<t}, \mathbf{x}, \theta) \tag{1}$$

Where $n$ is the number of prompts considered, $m_i$ is the target length after adding the $i$-th prompt.

### A.5 Robustness to Language Pair Distance

Translations between different language pairs contain different variances. Here we aim to measure how the performance of a metric will change considering the distance between a language pair. We use language vectors to measure the distance between two languages [1], and consider 6 distances, which are *syntactic*, *geographic*, *phonological*, *genetic*, *inventory* and *featural* distances. We plot the Pearson correlation heatmap in Fig. 1. We observe that the correlation doesn't change much w.r.t. different distances across metrics. And the results show that all metrics have a significant correlation with *genetic* distance. This indicates that metrics are good at measuring translation quality from genetically different languages. This may be because the translation from similar languages is easier than dissimilar languages, making translation systems less distinguishable.

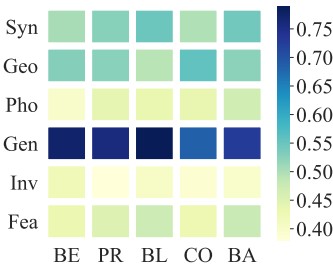

Figure 1: Pearson correlation between language pair distance and correlation with human metrics.