# OpenReview forum: "BARTScore: Evaluating Generated Text as Text Generation"
_NeurIPS.cc/2021/Conference — NeurIPS 2021 Poster_

### Official Review · Reviewer_rF8J · 2021-06-28

**Rating:** 8
**Confidence:** 5

**Summary:**

This paper introduces BARTScore, a new metric for generation based on the BART model. BARTScore frames the evaluation of generated text as a text generation problem. Additionally, the authors show prompts can be used to improve the metrics. BARTScore is evaluated on various generation tasks, including summarization, machine translation (MT) and data-to-text. The results are convincing: BARTScore and its variants outperform unsupervised metrics in nearly all cases and are comparable to the best supervised metrics on MT.

**Limitations And Societal Impact:**

Do not think they are issues here.

**Main Review:**

This paper introduces BARTScore, a new metric for generation based on the BART model. BARTScore frames the evaluation of generated text as a text generation problem. In practice, this means computing conditional probabilities under a (fine-tuned) BART model. For instance, a precision score can be obtained by computing p(h|r), the probability of generating the hypothesis given a reference text. Similarly, faithfulness can be computed as p(h|s) (generating the hypothesis given the source document) and recall can be computed as p(r|h).

BARTScore is unsupervised, in the sense that it does not require human judgements scores. However, fine-tuning the BART model on task-relevant datasets improves its correlation with human judgements. The authors experiment with tuning on CNN/DailyMail and tuning on CNN then ParaBank2.  This puts BARTScore in between fully rule-based metrics (e.g: n-gram overlap such as BLEU/ROUGE), and metrics directly fine-tuned to replicate human judgements (e.g BLEURT).

The authors show that adding prompts to BARTScore can further help its correlation with human performance. This is an interesting finding in itself and in line with recent findings on the effectiveness of prompting for large LMs.

BARTScore is evaluated on various generation tasks, including summarization, machine translation (MT) and data-to-text. The results are convincing: BARTScore and its variants outperform unsupervised metrics in nearly all cases and are comparable to the best supervised metrics on MT.

Pros:
- BARTScore is highly correlated with human judgments in a variety of generation settings. It also does not require pre existing human judgements. In addition, significance testing is performed, a welcome addition that confirms the usefulness of BARTScores.
- One possible caveat is that there are many BARTScores (4 metrics * 4 models tested), and score selection can be an issue when there are no dev set human judgements available. I would like to see this mentioned, however, I do not believe this to be a dealbreaker: 1/ The metric is chosen depending on the task, not selected 2/ The BARTScore variant that performs best for each task often makes intuitive sense (fine-tuned on paraphrasing for MT, on summarization for summarization) 3/ On many tasks, several BARTScore variants are ahead of other metrics.
- The idea of using prompting to make model-based metrics more effective is interesting, and shows great results here. I believe this is a worthwhile contribution per se. It will surely lead to other papers exploring this, as the idea can be pushed further (e.g: continuous prompts, etc.).
- The paper is very well-written. The Preliminaries section in particular is great and should be required reading for researchers focused on generation metrics.
- The code for BARTScore is already published, with a clear README. There is also a leaderboard for metric evaluation. Both of those will be very valuable to the community.
- The analysis, though short, has interesting findings.

Cons:
- It would be great to see some analysis on the impact of model choice to understand the generality of BARTScore. In particular, I am curious about:
  1.  How well does this work if BART is replaced with another, worse-performing model? Using older models to evaluate the quality of more recent ones can lead to issues as shown in BLEURT. I do not believe 4.4.1 fully addresses this given many evaluated models are either BART or older.
  2.  Are the scores improved on some tasks due to the base metric model being the same as the models evaluated (e.g Factuality evaluation of BART based outputs)?

~~~~

Author response edit: I still feel like this paper deserves the high grade I assigned it. Here is my rationale for sticking with a higher grade than my fellow reviewers:

- Many of the concerns highlighted by my fellow reviewers are interesting research questions themselves that apply more generally to learned/co-evolved evaluation methods (how much does it help to use a metric based on the same model being tested? tuned in the same way? trained in a seq-to-seq vs BERT like setup? what new biases do learned metrics introduce? What is the role of human evals in the co-evolution of metrics and models?). Although this paper does not answer all these questions satisfactorily, this cannot be expected of a single paper and the code release and result repository will prove fruitful for such studies.

- The prompt idea is very interesting in itself, something which has not been very emphasized in other reviews (aside from bvYP).

Consequently, and after reading the other reviews and response, I have increased my confidence score.

The main recommendation I have is that the authors mention the metric selection issue more explicitly (Weakness 1 of Rev TKFn, which I agree at least partially with cf my caveat).


**Time Spent Reviewing:**

3.5

---

> ### Author Response · Authors · 2021-08-10
> **Authors' Response**
>
> **Cons 1: "How well does this work if BART is replaced with another ..."**
>
> **A1:**
> Thank you, this is a good point. As shown in the Appendix, replacing BART with another worse-performing model like T5 results in worse correlation with human judgements. Also, please see the third response to reviewer 1 where we argue that metrics should evolve together with systems.
>
>
> **Cons 2: "Are the scores improved on some tasks due to the base metric model being the same as the models evaluated ... "**
>
> **A2:**
> This is a good point, please see the second response to reviewer 3 where we discuss in detail.

---

### Official Review · Reviewer_TKFn · 2021-07-13

**Rating:** 6
**Confidence:** 4

**Summary:**

This paper proposes BARTScore, a method to evaluate NLG system outputs by using a NLG system's output likelihood score, in particular different variants of BART. By changing what BARTScore conditions on and what it must score, as well as the prompts which are fed to the BARTScore model, variants of the evaluation method can be derived (faithfulness, precision, recall, F), and be adapted to measure different generation qualities such as coverage, coherence, factuality, fluency, informativeness, and adequacy. Experiments are run on a number of NLG tasks including MT, summarization, factuality, and data-to-text, where higher correlations are found between versions of BARTScore and human judgments. There are also several extra analytical experiments on MT focusing on issues such as reference length and the choice of prompts.

**Limitations And Societal Impact:**

There isn't any deep discussion of limitations or societal impacts of their work. There is some discussion of how a research community that adopts BARTScore might lead to a research cycle where the evaluation method and the system would co-evolve. It is unclear to me whether this would be a positive development.

**Main Review:**

This paper proposes a new method to evaluate NLG system outputs based on using scores derived from finetuned BART-based conditional language models. There are a large number of experiments to evaluate the correlation of the method to human judgments. The paper is clearly written and easy to understand, and the contribution should be significant to the NLG community if it is adopted.

Strengths:
 - The proposed method is simple and flexible. I appreciated the different methods of applying BARTScore that consider the relation between the source text, the reference output, and the reference output, which can be adapted to the quality of the text being evaluated.
 - BARTScore achieves good correlations on a range of NLG datasets in a range of NLG tasks.

Nevertheless, there are a few issues with the paper which are concerning.

Weaknesses:
1 - First, the reporting of the results and the comparisons to previous methods are unfair and potentially misleading. For some of the results, notably Table 4 for summarization evaluation, BartScore gets an advantage in being able to select different variants of itself for the different subtasks, as described by the paper. This is in comparison to other methods which must stick to one variant, such as ROUGE-1 or ROUGE-L. Second, many of the previous measures presented are not designed to measure the stated qualities. For example, ROUGE-1 was never meant to measure coherence, or indeed factuality, or fluency. Thus the table is misleading and overstates the advantage that BARTScore variants have over previous methods.

2 -  On a different level, even if we take the better correlations at face value, I am concerned that BARTScore would introduce biases in unpredictable ways. For example, could it privilege models that are more BART-like over NLG models that are not based on pre-trained LMs? I would be interested in seeing whether the correlations hold within AND between model classes. e.g., what if they were applied to a dataset with a combination of neural and pre-neural MT or summarization systems? What if BARTScore were applied to reference outputs themselves?

3 - Another interesting analysis to perform would be to examine and evaluate the output of decoding directly from the probability distribution of BARTScore. I would expect BARTScore to break down when evaluating these extrema or similar-looking sentences, and a question to ask is how do we know if we encounter such cases?

---
Edit:
The response and the proposed revisions addressed Weaknesses 2 and 3 above satisfactorily, but we still disagree regarding Weakness 1. I've updated my score to reflect this.


**Time Spent Reviewing:**

2

---

> ### Author Response · Authors · 2021-08-10
> **Authors' Response**
>
> **Weaknesses 1: "First, the reporting of the results and the comparisons to previous methods are unfair and potentially misleading ..."**
>
> **Response 1:**
> In fact, many other metrics also have variants. For example, ROUGE scores also have precision, recall and F score; BERTScore and others are similar. When conducting the experiments in Table 4, we select the most effective variant for all metrics. For example, for the REALSumm datasets, the human-annotated pyramid score is a recall-based metric and consequently we found that the recall version of all metrics outperform their other variants. Because of this, we adopted the recall version for all metrics. For other summarization datasets, even though we list some variants of BARTScore, the src→hypo version is the highest-performing one as stated in the paper and in the paragraph "Selection of BARTScore Usage." in 4.2.2. It's true that other metrics, such as ROUGE, are not meant to measure coherence. However, we would argue that the fact that BARTScore can perform evaluation from different perspectives is one of its major advantages, and given that we have used the most appropriate variant of each baseline metric, we believe this is a fair comparison.
>
>
> **Weakness 2: "On a different level, even if we take the better correlations at face value ..."**
>
> **Response 2:**
> This is a good point, so we decided to dig a bit deeper. Bias would indicate that the scores are too high or too low compared to the scores they are given by human annotators. Therefore, to see whether such biases exist, we inspected the rank differences given by human annotators and BARTScore on the REALSumm dataset where 24 systems are considered, including both abstractive models and extractive models as well as models based on pre-trained models and models that are trained from scratch. Below are the systems that have rank differences over 2.
>    * 'pnbert_out_lstm_pn_rl [1]   (Real rank: 4, Predicted rank: 10)
>    * 'pnbert_out_bert_tf_sl [1]  (Real rank: 6, Predicted rank: 11)
>    * 'pnbert_out_bert_tf_pn [1]  (Real rank: 7, Predicted rank: 13)
>    * 'neusumm_out [2]  (Real rank: 10, Predicted rank: 4)
>    * 'heter_graph_out [3]  (Real rank: 11, Predicted rank: 8)
>    * 'banditsumm_out [4]  (Real rank: 12, Predicted rank: 15)
>    * 'unilm_out_v1 [5]  (Real rank: 15, Predicted rank: 5)
>
> 1. Zhong M, Liu P, Wang D, et al. Searching for effective neural extractive summarization: What works and what's next[J]. arXiv preprint arXiv:1907.03491, 2019.
> 2. Zhou Q, Yang N, Wei F, et al. Neural document summarization by jointly learning to score and select sentences[J]. arXiv preprint arXiv:1807.02305, 2018.
> 3. Wang D, Liu P, Zheng Y, et al. Heterogeneous graph neural networks for extractive document summarization[J]. arXiv preprint arXiv:2004.12393, 2020.
> 4. Dong Y, Shen Y, Crawford E, et al. Banditsum: Extractive summarization as a contextual bandit[J]. arXiv preprint arXiv:1809.09672, 2018.
> 5. Dong L, Yang N, Wang W, et al. Unified language model pre-training for natural language understanding and generation[J]. arXiv preprint arXiv:1905.03197, 2019.
>
> 6/7 of those systems are extractive systems and some of them are overestimated while others are underestimated. This indicates that BARTScore is less good at distinguishing the quality of extractive systems and much better at distinguishing the quality of abstractive systems. Thanks for suggesting this, we will try to further refine the analysis, add comparisons with other evaluation metrics, and add this discussion into our revised paper.
>
>
> **Weakness 3: " Another interesting analysis to perform would be to examine and evaluate ... "**
>
> **Response 3:**
> Thanks, this is a great idea. We decoded 20 outputs from a source text and calculated their BARTScore. They all look pretty similar and we select the highest-scoring one and lowest-scoring one. Below are the results.
>
> Source text: For the first time in eight years, a TV legend returned to doing what he does best. Contestants told to "come on down!" on the April 1 edition of "The Price Is Right" encountered not host Drew Carey but another familiar face in charge of the proceedings. Instead, there was Bob Barker, who hosted the TV game show for 35 years before stepping down in 2007. Looking spry at 91, Barker handled the first price-guessing game of the show, the classic "Lucky Seven," before turning hosting duties over to Carey, who finished up. Despite being away from the show for most of the past eight years, Barker didn't seem to miss a beat.
>
> Decoded texts:
>    1) Bob Barker returned to "The Price Is Right" for the first time in eight years. He hosted the TV game show for 35 years before stepping down in 2007. Barker handled the first price-guessing game of the show, the classic "Lucky Seven" (BARTScore: -0.38535189628601074)
>    2) Bob Barker returns to "The Price Is Right" for the first time in eight years. The 91-year-old hosted the show for 35 years before stepping down in 2007. He handled the first price-guessing game of the show, the classic "Lucky Seven." (BARTScore: -0.4631081819534302)
>
> Observation: although the two look pretty similar, the tense in the first one is more consistent than the second one. Therefore, the BARTScores given to those two sentences seem reasonable. We will add this discussion in our revised version.

---

> > ### Comment · Reviewer_TKFn · 2021-08-17
> > **Reply to response**
> >
> > Thanks for your response. I think your proposed revisions regarding Weaknesses 2 and 3 would adequately address my concerns, and I'll adjust my score to reflect this.
> >
> > I still disagree about Weakness 1. I don't think the way Table 4 compares the performance of different methods is fair, as BARTScore has more dimensions of variations and ROUGE and other baseline evaluations measures were not designed to measure many of the tested criteria. I think at a minimum, the "Avg." column should be removed, and there should be prominently displayed caveats in the Table caption about what the baselines were originally developed to measure in order to not mislead practitioners.

---

### Official Review · Reviewer_3xwH · 2021-07-16

**Rating:** 8
**Confidence:** 3

**Summary:**

This paper tries to propose a new generation-based BARTScore to evaluate text generation problems. The authors want to fill the gap between pretrainining objectives and the down stream feature extractors.  The main idea is that BART can achieve better score when the generation results are better. The BART Score is an unsupervised metric and have better correlation with different perspectives. The paper also check the influence of different textual prompts and fine-tuning process. The paper conducts experiments from 7 perspectives on 16 datasets.

**Limitations And Societal Impact:**

The paper describes some limitations in section 5. The paper also provides future work directions in that section.

**Main Review:**

Strengths:
1. The paper gives a pretty solid literature survey. The paper first formally defined gold-standard human evaluation with 7 different perspectives. The paper then characterizes differences in task formulation and evaluation perspectives. The baselines are all state-of-the-art or widely used in the generation/summarization/translation tasks. The paper also provides code and sample scripts for users to check. The paper also provides pre-trained model selection and prompt set in the supplements.

2. The idea to use seq2seq generation tasks is pretty novel. The paper provides an unsupervised way to evaluate generation results with the transformer-based pre-trained seq2seq model. The paper provides four ways to apply their model according to different situations. The experiments are pretty comprehensive. The paper compares BARTscore with supervised and unsupervised metrics by using multiple evaluations. The paper also provides insightful analysis about reference length and differences prompt.

Weaknesses:
1. In the experiments for translation Table 3 and Figure 2, I don't see a clear difference between COMET and BARTSCORE. The COMET has close or higher scores compared to BARTSCORE. It would be better for authors to clarify the detailed advantages in the paper. The authors seem to only focus on the automatic metrics with quantitative evaluation. I would like to see more qualitative evaluation with detailed examples like section 4.4.2.

2. In the paper, it is not clear the specific BART model. The paper might be better to state that they use BART-large-CNN as shown in the code. Also, I'm curious about the reason behind the bad performance of T5 and PEGASUS. Moreover, it seems that T5-base-CNN only has about 200M parameters. However, its performance is comparable with BART-CNN which has 400M parameters. I'm wondering why authors choose BART-CNN instead of T5-base-cnn. The paper also fails to specify which subset of prompt they uses for each task.

**Time Spent Reviewing:**

4

---

> ### Author Response · Authors · 2021-08-10
> **Authors' Response**
>
> **Weakness 1: "In the experiments for translation Table 3 and Figure 2 ..."**
>
> **Response 1:**
> Certainly, we are happy to clarify the advantages both here and in a paper revision. Most importantly, COMET requires human judgements to train which are costly to collect in large amounts, whereas BARTScore does not. Because of this, BARTScore is applicable to a wide variety of tasks and evaluation perspectives, whereas COMET’s scope is more limited, mostly only to MT. Despite this fact, BARTScore is still very competitive with COMET on the MT task where COMET has been trained with extensive human judgements. We also believe that BARTScore could be further trained on human judgements data using ranking losses, and given that BARTScore is already competitive when trained entirely unsupervised, we hypothesize that it might surpass COMET in this case (although this would obviously need to be experimentally verified). Separately from this, we are happy to add more examples such as the ones in 4.4.2.
>
>
> **Weakness 2: "In the paper, it is not clear the specific BART model ..."**
>
> **Response 2:**
> The reason for the bad performance of T5 and PEGASUS could be the different pre-training objectives. Since we used MT datasets for model selection, it is mainly the model’s ability to identify paraphrases that is being tested. We hypothesize that BART's denoising pre-training scheme may have learned to paraphrase better since BART aims to recover the whole input text while T5 only calculates the loss over the corrupted part.
>
> We use BART-CNN instead of T5-base-cnn because BART-CNN performs better than T5-base-CNN across all language pairs.
>
> We apologize for the lack of clarity on the prompts, which we will rectify in a revised version. We divide the prompt set into two subsets, one for src-->hypo, one for hypo<-->ref. Except for MT datasets where we only used one prompt, we used all the prompts in the corresponding subset for a BARTScore version for other datasets. We will make this more explicit in our paper.

---

> > ### Comment · Reviewer_3xwH · 2021-09-02
> > **Thanks for your response**
> >
> > Thank you very much for your response. The author's response has addressed my concern. I have updated my scores.

---

### Official Review · Reviewer_bvYP · 2021-07-26

**Rating:** 6
**Confidence:** 4

**Summary:**

This paper proposes a different paradigm of evaluation of text generation by modeling evaluation as a text generation problem. This is built on top of the BART text generation algorithm, hence, the metric is called BARTScore. Similar to BERTScore, which is based on BERT, the metric proposed in this paper requires no training (with the reference human judgments) as the pre-trained BART model is used. BARTScore considers various aspects of matching generated text with reference (like factuality, coherence, etc.). In addition, BARTScore is shown to improve with using textual prompts and with fine-tuning on downstream domain tasks. BARTScore shows promising results across the board and fine-grained analysis shows effectiveness across different perspectives.


**Limitations And Societal Impact:**

No, the authors have not addressed the concerns of moving targets in evaluation which can lead to significance biases in text generation systems.

**Main Review:**

## Strong points:
1. The paper is well-motivated and clear. The idea of a different paradigm for evaluation is interesting.
2. The premise of the idea is well-positioned against relevant prior work.
3. An important contribution is the addition of prompts which can lead to better evaluations.
4. A bunch of different standard metrics has been used to compare BARTScore with promising results.
5. The fine-grained analysis in Sec 4.4 offers some necessary insights.

## Questions:
1. The idea of using text generation as an evaluation of text generation is not entirely new (as the authors themselves compared with PRISM). So, the BARTScore formulation and the finer points of differences with PRISM seem incremental.
2. It is not clear why the authors resorted to weighing each token equally (is it experimentally validated against IDF?).
3. There is a concern for this kind of metric which is dependent upon existing text generation like BART. The paper also talks about trying out T5 and PEGASUS. The metric which is dependent upon a pre-trained model is like a moving target with no clear grounding of evaluations. Let's say a future model BART++ comes up, how do you decide whether it is better to have BARTScore or BART++Score. It's also not clear how to establish that BART++ is better than BART in text generation when the evaluation metrics are moving targets. The authors advocated this paradigm of evaluation and text-gen systems co-evolving together but that can introduce significant biases without the proper grounding of evaluations.
4. Fine-tuning on downstream tasks is not surprising and is a positive outcome. However, the concern remains because the evaluation is now dependent on the dataset being fine-tuned on, which aggravates the need for having a non-moving gold standard evaluation as highlighted above.
5. Section 4.1.2 definitions seem lifted from Wikipedia word-by-word even though some other articles have been cited.
6. The paper gives a feel of being hurriedly written, and a lot of work is needed to make it a coherent and fun read.
7. A bunch of different datasets has been considered for comparison covering various aspects like coherence, consistency, informativeness, etc. However, for Factuality and Data-to-text datasets, the number of samples for evaluation is in 100s, the scale of which is very less to lead to a strong conclusion about the effectiveness of BARTScore over others.
8. In Sec 4.2.2, the subsection "Selection of BARTScore Usage", there are intuitions about the preference of the kind of BARTScore used. Have they been experimentally validated as well?
9. Typo: Line 159: 'encouraged' --> 'encourage'.

## Originality: 5/10

## Quality: 4/10

## Clarity: 6/10

## Significance: 4/10



**Time Spent Reviewing:**

10

---

> ### Author Response · Authors · 2021-08-10
> **Authors' Response**
>
> **Q1: "The idea of using text generation ..."**
>
> **A1:**
> BARTScore’s formulation is different from that of PRISM according to four perspectives:
> 1. PRISM formulates MT evaluation as paraphrasing, which limits its applicability (e.g. it cannot easily be applied to factuality evaluation). However, BARTScore’s text generation formulation doesn’t require that texts be paraphrases, which results in a wider range of potential applications.
> 2. PRISM trains its Transformer model from scratch by using parallel translation data, while BARTScore takes advantage of existing pre-trained models.
> 3. Because of the two above points, BARTScore can outperform PRISM by a large margin in all three tasks we have performed.
> 4. Besides, we incorporate textual prompts to further enhance the performance, which hasn’t been explored in PRISM.
>
>
> **Q2: "It is not clear why the authors resorted to weighing ... "**
>
> **A2:**
> We tried many different weighting schemes including
> 1. Use the IDF of tokens
> 2. Ignore the stop words
> 3. Use the prior probability of tokens (directly generate the target sentence with an empty source and get the generation probability of each token)
> 4. Use the reciprocal of the prior probability of tokens
>
> None of those outperform the uniform weighting scheme, so we only used uniform weighting in our paper. Nevertheless, thanks for pointing this out, and we will modify the paper to both make the fact that we are using uniform weighting clear, and briefly describe the other attempts as well.
>
>
> **Q3: "There is a concern for this kind of metric which is dependent ... "**
>
> **A3:**
> We totally understand the concern about lack of comparability and moving targets, this is an important concern. However, we would counter-argue:
>
> 1. Even simple metrics such as BLEU and ROUGE have widely-known issues with comparability due to tokenization or other issues, and thus require standardization through libraries such as SacreBLEU. We believe that this sort of standardization in software and reporting will be both possible and necessary in all future metrics. The BARTScore library will support this, and we will encourage users to report the version of BARTScore they use in papers.
>
> 2. Better pre-trained models will undoubtedly result in better evaluation. In fact BERTScore has an extensive evaluation of hundreds of pre-trained models demonstrating this on their github repository. We would argue that we are now in the age where not only models, but also evaluation, must evolve together with our pre-trained models. Of course, these evolving metrics must be appropriately meta-evaluated with respect to human judgements (which would help with the decision between BARTScore and BART++Score), and one of our contributions is also an extensive meta-evaluation library to help out with this.
>
>
> **Q4: "Fine-tuning on downstream tasks ... "**
>
> **A4:**
> We believe our answer above largely addresses this concern as well. In addition, using supervised objectives in addition to unsupervised objectives to learn the representations used in evaluation is not unique to our work by any means. For example, MoverScore uses BERT fine-tuned on MNLI dataset, similarly to how we do for BARTScore.
>
>
> **Q5: "Section 4.1.2 definitions seem lifted from ..."**
>
> **A5:**
> Thanks for pointing this out, and we will re-write the definitions in our own words.
>
>
> **Q6: "The paper gives a feel of being hurriedly written ..."**
>
> **A6:**
> Thank you for pointing this out, we will re-read the paper carefully and improve the writing wherever possible. Any additional suggestions are highly welcome of course.
>
>
> **Q7: "A bunch of different datasets ..."**
>
> **A7:**
> Even though the datasets for factuality and data-to-text are relatively small, we have performed significance tests in our paper (see 4.2.2 last paragraph). According to our significance tests, BARTScore is indeed better than other metrics on those datasets. In a way, it could be said that significant results on these smaller datasets are perhaps even stronger evidence than those on larger datasets (given the weaker underlying statistical power).
>
>
> **Q8: "In Sec 4.2.2, the subsection "Selection of BARTScore Usage" ..."**
>
> **A8:**
>
> Summarization & factuality
> * For REALSumm, we use the recall version since pyramid recall is a recall based human metric. This is validated in our experiments and we found that the recall version of all metrics (if they exist) achieve better correlation with pyramid scores than other variants like precision. This is in line with our first suggestion in “Selection of BARTScore Usage”.
> For other datasets, we observe that the src-->hypo version pre-trained on CNNDM performs the best, compared to ref<-->hypo version. This is the reason for our second suggestion in “Selection of BARTScore Usage”.
>
> MT & Data-to-text
> * We do not have a pre-trained src-->hypo variant for these tasks yet because the source text and hypothesis text are in different languages or different modalities. Therefore, we only considered the hypo↔ref version. And this variant works already reasonably well, which is why we have our third suggestion in “Selection of BARTScore Usage”.
>
>
> **Q9: "Typo: Line 159: 'encouraged' --> 'encourage'."**
>
> **A9:**
> Thanks for pointing out this typo, we will modify it in our paper.

---

### Decision · Program_Chairs · 2021-09-27

**Decision:**

Accept (Poster)

**Comment:**

This paper proposes a framework for evaluation of text generation (BARTScore) by modeling evaluation as a text generation problem, built on top of the BART text generation algorithm. Similar to BERTScore, which is based on BERT, the metric proposed in this paper requires no training. BARTScore is also shown to improve with using textual prompts and with fine-tuning on downstream domain tasks, showing good results across the board for evaluating various generation tasks including MT and summarization, approaching evaluation metrics such as COMET which are supervised on human assessments.

All reviewers feel positively about this paper. While the underlying idea is not particularly creative, this is a solid paper which proposes a simple and effective approach for evaluation of text generation with convincing results and which is likely to be impactful.